# Modalities Contribute Unequally: Enhancing Medical Multi-modal Learning through Adaptive Modality Token Re-balancing

Jie Peng [* 1]   Jenna L. Ballard [* 2]   Mohan Zhang [3]   Sukwon Yun [3]   Jiayi Xin [2]   Qi Long [2]   Yanyong Zhang [1]
Tianlong Chen [3]

## Abstract

Medical multi-modal learning requires an effective fusion capability of various heterogeneous modalities. One vital challenge is how to effectively fuse modalities when their data quality varies across different modalities and patients. For example, in the TCGA benchmark, the performance of the same modality can differ between types of cancer. Moreover, data collected at different times, locations, and with varying reagents can introduce inter-modal data quality differences (*i.e.*, **Modality Batch Effect**). In response, we propose **A**daptive **M**odality Token Re-Balan**C**ing (AMC), a novel top-down dynamic multi-modal fusion approach. The core of AMC is to quantify the significance of each modality (Top) and then fuse them according to the modality importance (Down). Specifically, we assess the quality of each input modality and then replace uninformative tokens with inter-modal tokens accordingly. The more important a modality is, the more informative tokens are retained from that modality. The self-attention will further integrate these mixed tokens to fuse multimodal knowledge. Comprehensive experiments on both medical and general multi-modal datasets demonstrate the effectiveness and generalizability of AMC. Code is available at https://github.com/PengJieb/amc.

## 1. Introduction

Multi-modal learning has advanced medical analysis by integrating diverse data sources (e.g., clinical, imaging, ge-

*Equal contribution [1]University of Science and Technology of China [2]University of Pennsylvania [3]University of North Carolina at Chapel Hill. Correspondence to: Yanyong Zhang <yanyongz@ustc.edu.cn>.

*Proceedings of the 42$^{nd}$ International Conference on Machine Learning*, Vancouver, Canada. PMLR 267, 2025. Copyright 2025 by the author(s).

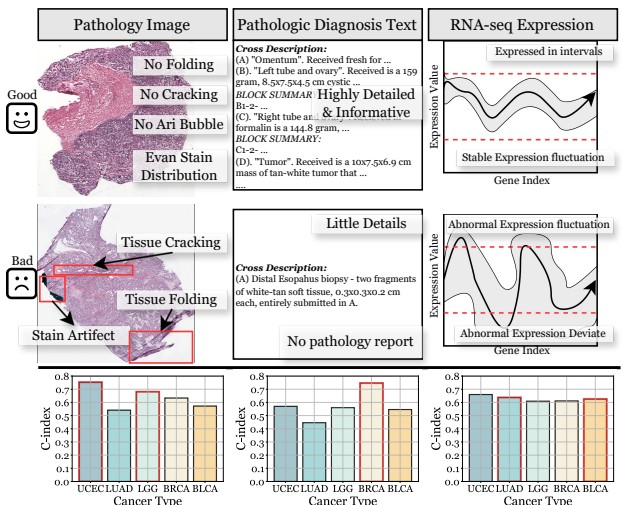

*Figure 1.* The top part of the figure illustrates how the quality of different modalities can vary dynamically. The bottom part shows how variations within the same modality across different cancer types can directly impact final task performance. The red box indicates the best performance across these three modalities.

netic) (Han et al., 2024; Yun et al., 2024a; Mobadersany et al., 2018; Yan et al., 2021; Yap et al., 2018; Boehm et al., 2022a; Vanguri et al., 2022; Zhou et al., 2023; Lipkova et al., 2022; Chen et al., 2022a; Steyaert et al., 2023; Boehm et al., 2022b; Im et al., 2023; Chen et al., 2024b). These data provide a comprehensive view of patient health, enabling personalized treatments and enhancing diagnosis and prediction across medical applications.

One vital challenge in multi-modal medical learning is effective multi-modal fusion when data quality varies across different modalities and patients (see Figure 1) (Liu et al., 2024a; Chen et al., 2024a; Sparring et al., 2018; Ghalavand et al., 2024). Most previous fusion methods are designed for visual, text, and audio data (Lin et al., 2024b; Sun et al., 2021; Joze et al., 2020; Cao et al., 2023), but they do not address the specific needs of the medical domain. Medical data involves more heterogeneous modalities, such as genomics, neuroimaging, and biofluid biomarkers, which require specialized fusion techniques.

Prior dynamic fusion (Wang et al., 2022b; Sun et al., 2021;

Joze et al., 2020; Cao et al., 2023) methods use mechanisms like self-attention (Yun et al., 2024a; Han et al., 2024), cross-attention (Wu et al., 2023), or gating networks (Rahman et al., 2020) to implicitly decide each modality's contribution and fuse them accordingly. Recently, in the era of multi-modal large language models (MLLMs) (Lin et al., 2024b; Liu et al., 2025; Guo et al., 2025; Liu et al., 2023b; 2024b; Li et al., 2023a), approaches like LLaVA (Liu et al., 2023b; 2024b) project other modalities into the language token space, utilizing self-attention to integrate all modalities. This has increased interest in improving self-attention for multi-modal fusion (Wu et al., 2024; Zhao et al., 2023). However, vanilla self-attention can be problematic, as it often assigns diluted attention weights to irrelevant contexts, leading to sub-optimal performance (Wang et al., 2022b; Ye et al., 2024). A possible solution is token fusion (Wang et al., 2022b), which replaces uninformative tokens with inter-modal tokens to better integrate information from various modalities for dynamic fusion. Token fusion has shown effectiveness and efficiency in visual fusion tasks. However, it focuses on alignment-aware fusion, requiring explicit relationships between modalities, such as shared pixels or 3D coordinates. This requirement limits its application in the medical domain, where such relationships are difficult to ascertain, like linking a sub-region of a pathology image with specific genes.

In response, we start with the perspective that **modalities contribute unequally**. Specifically, we propose a top-down dynamic multi-modal fusion method called the Adaptive Modality Token Re-Balancing method (AMC), which dynamically balances modality contributions via self-attention. Unlike previous methods that learn modality contributions implicitly, we emphasize that **explicitly recognizing unequal contributions of modalities could be more beneficial**. Our AMC method works in two main steps. First (Top), it identifies the importance of each modality. We calculate this using attention distribution. This step helps determine how many tokens from each modality should be replaced with tokens from other modalities. This explicit calculation makes each modality's contribution interpretable, which is crucial in the AI4Medical field (Duan et al., 2024; Wang et al., 2024). Second (Down), we replace tokens with lower scores with tokens from other modalities, based on the modality's importance. The token score, also derived from attention distribution, guides this token replacement process. This approach ensures that we prune uninformative tokens and utilize informative tokens from other modalities for fusion. In our model design, we use a single transformer branch to process all modalities. We introduce specific modules within the transformer block to enhance parameter efficiency and fusion quality, supporting the AMC. These include an improved self-attention design, a sparse mixture-of-experts (SMoE) module, and a two-level contrastive learning loss.

We demonstrate the effectiveness of AMC through extensive experiments on several real-world datasets, including the MIMIC-IV dataset, the Alzheimer's Disease Neuroimaging Initiative (ADNI) dataset, and a subset of the TCGA benchmark covering five different cancer types. The results confirm the robustness of AMC across diverse medical scenarios. The contributions of this work are as follows:

- We introduce a novel top-down dynamic fusion mechanism that adaptively integrates multi-modal information, addressing the variation in data quality across different modalities and patients in various medical domains.
- AMC features a novel token fusion approach, including attention-based modality importance calculation, token score assignment, token replacement policy, and a customized transformer block design.
- Extensive experiments on real-world datasets, including ADNI, MIMIC-IV, and TCGA, demonstrate the consistent and robust performance of AMC in handling variations in data quality.

## 2. Related Work

**Multi-modal Learning with Transformers.** Recently, Transformer-based Multimodal learning has achieved substantial progress(Liu et al., 2023a; Li et al., 2023b; Lin et al., 2024a; Dai et al., 2023), due to their unique advantages and scalability in modeling different modalities, such as language, image, audio, video, etc. Inspired by the great success of Transformer-based pretraining in the NLP domain(Touvron et al., 2023; Devlin et al., 2019; Brown et al., 2020; Bai et al., 2023; DeepSeek-AI et al., 2024), a series of works(Dosovitskiy et al., 2021; Sun et al., 2019; Radford et al., 2021; Li et al., 2022a; Lu et al., 2019; Chen et al., 2020) have demonstrated that if pre-trained on large-scale multi-modal datasets, Transformer-based models outperform other domain-specific models in a wide range of multi-modal tasks, and more importantly, achieve remarkable zero-shot generalization ability. For example, (Liu et al., 2024c) proposes a tight fusion solution to fuse language and vision modalities, and develops an open-set object detector that can detect arbitrary objects with language inputs such as category names or referring expressions.

**Dynamic Multi-modal Fusion.** In multi-modal learning, the importance of modalities can vary based on the task and input data quality. Thus, dynamic fusion methods that adapt to these changes are crucial (Zhang et al., 2024; Guan et al., 2019; Han et al., 2020; Tian et al., 2020; Chen et al., 2022b; Sun et al., 2021; Hazarika et al., 2018; Joze et al., 2020; Cao et al., 2023; Yun et al., 2024a). TensorFusion (Zadeh et al., 2017) integrates all modalities by using a tensor fusion network to handle both intra-modality and inter-modality dynamics. FlexMoE (Yun et al., 2024a) and

LiMoE (Mustafa et al., 2022) use self-attention with sparse mixture-of-experts to adaptively integrate multiple modalities and reduce interference. FuseMoE (Han et al., 2024) and MAGGate (Rahman et al., 2020) apply gating networks for dynamic fusion. MUSE (Wu et al., 2023) uses graph networks and contrastive learning to fuse modalities effectively for predictions. Token fusion (Wang et al., 2022b) minimizes computation and prevents attention from focusing on irrelevant contexts by using inter-modal tokens. Despite these advancements, most methods focus on traditional scenarios like visual detection, semantic segmentation, and audio-visual tasks. Evaluating each modality's contribution and effectively fusing heterogeneous modalities is still a challenge, especially in medical multi-modal tasks.

## 3. Preliminaries and Notations

**Token Fusion.** Token fusion (Wang et al., 2022b) is originally designed for fusing vision transformers with homogeneous or heterogeneous modalities. After each transformer layer, it employ addition feed-forward MLP layer as scoring layer $s_m(\cdot)$ to dynamically predicts the importance of tokens for the $m$-th modality. Given input multi-modal tokens $\{\mathbf{x}_m\}_{m=1}^M$ and the inter-modality token is well aligned[1], the token fusion of $n$-the token is defined as:

$$
\begin{aligned}
\mathbf{x}_m[n] = \; & \mathbf{x}_m[n] \odot \mathbb{I}_{s(\mathbf{x}_m[n]) \geq \theta} \\
& + \sum_{m' \neq m}^{M} \mathbf{a}_{m'}(m)[n] \odot h(\mathbf{x}_{m'}[n]) \odot \mathbb{I}_{s(\mathbf{x}_m[n]) < \theta}
\end{aligned}, \quad (1)
$$

where $M$ is the number of all modalities, $N$ is the number of tokens of each modality, and $\mathbb{I}$ is an indicator asserting the subscript condition, therefore it outputs a mask tensor $\{0, 1\}^N$. The $\mathbf{x}_m[n]$ denotes the $n$-th token of $m$-th modality. The parameter $\theta$ is a small threshold, and the operator $\odot$ resents the element-wise multiplication. The $\mathbf{a}_{m'}(m) \in \{0, 1\}^N$, and $\mathbf{a}_{m'}(m)[n] = 1$ indicators the $n$-th token of the $m$-th modality can be substituted by the corresponding token of the $m'$-th modality, otherwise $\mathbf{a}_{m'}(m)[n] = 0$. When the number of modality $M = 2$, the $\mathbf{a}_{m'}(m)[n]$ is set to 1 by default. The $h(\cdot)$ is used to select tokens that correspond to the same things between modalities $m$ and $m'$. For homogeneous modalities, $h(\cdot)$ is an identity function. For heterogeneous modalities, $h(\cdot)$ is defined as a projection function that uses modality-relevant domain knowledge to map the input token to its equivalent in another modality.

However, the original token fusion has several drawbacks which limit it applicability: (1) It assumes an explicit mapping relationship between modalities, which is necessary to

map one modality $m$ into another modality $m'$ for replacement. In many medical tasks, such explicit relationships are difficult to establish. For example, the relationship between pathology images and genetic profiles is unclear, and there is no exact mapping between them. (2) The setting of the parameter $\theta$ requires additional effort. A reasonable value for $\theta$ varies across tasks. For instance, tasks in the original paper set $\theta = 0.01$. However, in our medical tasks, no tokens would be replaced with $\theta = 0.01$. (3) The values in $\mathbf{a}_{m'}(m)$ are randomly predefined. Consequently, if $\mathbf{a}_{m'}(m)[n] = 0$, the $n$-th token will never be replaced from $m'$ to $m$. It assumes that all modalities can capture the right information about the same objects, neglecting the heterogeneity and varying abilities of different modalities.

## 4. Top-down Dynamical Fusion: `AMC`

In this section, we introduce our overall network architecture (Section 4.1) and the proposed **A**daptive **M**odality Token Re-Balan**C**ing (`AMC`) method. `AMC` accounts for the varying contributions of each modality across instances. It comprises two key operations: **Modality Importance Calculation** (Section 4.2) and **Customized Token Fusion** (Section 4.3). The **Modality Importance Calculation** computes the importance of each modality using attention distribution. It dynamically identifies contribution differences between modalities. The **Customized Token Fusion** then replaces these tokens with inter-modal features based on their calculated importance. For each instance, we calculate the modality importance to ensure effective fusion. Finally, we introduce two key network designs that enhance token fusion quality (Section 4.4).

### 4.1. Overall Network Architecture

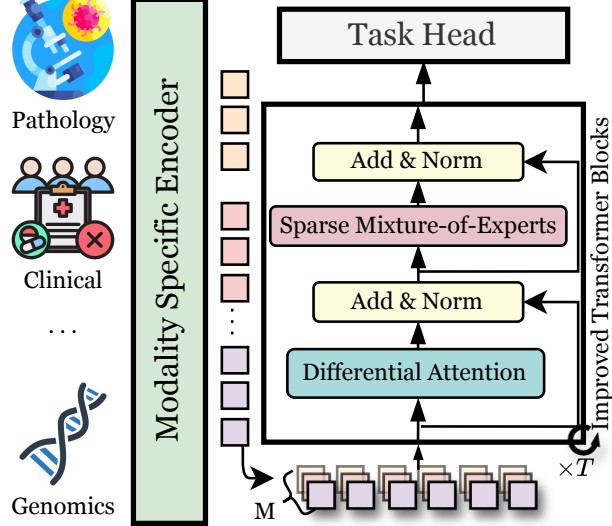

*Figure 2.* **Network Architecture Overview.**

As shown in Figure 2, our overall network architecture com-

---

[1]The $n$-the token $\mathbf{x}_m[n]$, $\mathbf{x}_{m'}[n]$ for modality $m$ and $m'$ are represent the same things in the physical world. For instance, $\mathbf{x}_m[n]$ and $\mathbf{x}_{m'}[n]$ denote the same object on corresponding RGB and depth images.

prises three components in turn: the modality-specific encoder, consecutive improved transformer blocks with AMC, and task head. ❶ The modality-specific encoder encodes modality $m$ into a sequence of tokens $\mathbf{x}_m \in \mathbb{R}^{N \times D}$, where $N$ is the number of tokens, and $D$ is the feature dimension. The structure of the modality-specific encoder relies on corresponding domain knowledge. For each task, we use Q-Former (Li et al., 2023c) in the modality-specific encoder to ensure the number of tokens $N$ is *the same across input modalities*. ❷ The improved transformer block replaces the self-attention module with the differential attention module (Ye et al., 2024) and substitutes the MLP layer with a Sparse Mixture-of-Experts (SMoE) layer. Subsequently, we concatenate multi-modal input $\{\mathbf{x}_m\}_m^M$ along the batch dimension, resulting in the transformer input $\alpha \in \mathbb{R}^{M \times N \times D}$, where $M$ is the number of modalities. After passing through the consecutive transformer layers, we split $\alpha$ back into $\{\mathbf{x}_m\}_m^M$. Each $\alpha_m$ undergoes average pooling, and the pooled outputs are concatenated and reshaped into a single feature vector. ❸ This vector is then fed into the task head, which employs a linear layer to produce the final prediction. To enhance modality fusion, we incorporate AMC between adjacent transformer layers. This approach facilitates the effective integration of information across modalities.

The computational increase from multi-modal input primarily arises from the modality-specific encoders. During fusion, the token sequence length remains the same as for a single modality input, $N$, but the batch size increases by a factor of $M$, the number of modalities. The computational complexity of concatenating all modalities into a longer sequence for the fusion is $\mathcal{O}((MN)^2)$. Using cross-attention for fusion, ensuring attention between every pair of modalities also has a complexity of $\mathcal{O}(\binom{M}{2}(N)^2) = \mathcal{O}((MN)^2)$. However, our computational complexity in the attention module is $\mathcal{O}(M(N)^2)$. The increased computational load from multiple modalities of our method can be handled by the parallel computing power of GPUs. Thus, with sufficient GPU power, the actual complexity in the attention module reduces from $\mathcal{O}(M(N)^2)$ to $\mathcal{O}(N^2)$.

### 4.2. Modality Importance Calculation

To obtain the importance of each input modality for subsequent customized token fusion. We utilize the attention distribution from the last transformer block, represented by the attention map $A = \mathbb{R}^{M \times N \times N}$. **First**, we compute the maximum value across the last dimension for each element, resulting in $\max_k(A_{m,n,k}) \in \mathbb{R}^{M \times N}$. This operation helps us capture the significance of each token. **Next**, we compute the average across the sequence dimension (the last dimension) of $\max_k(A_{m,n,k})$ to obtain the *statistical information* as the modality importance indicator. This yields the vector $\sigma \in \mathbb{R}^M$, where each element $\sigma[m]$ corresponds to the average of the maximum values for the $m$-th modality across

all sequences. **Last**, we normalize $\sigma$ and interpret as the modality importance $\mathbf{s} \in [0,1]^M$. The vector $\mathbf{s}$ emphasizes which modalities have the most consistent maximum values, indicating their relative importance. In summary, the modality importance $\mathbf{s} \in [0,1]^M$ is obtained as follows:

$$\sigma[m] = \mathtt{Mean}(\max_k(A_{m,n,k})[m,:])$$
$$\mathbf{s} = \frac{\sigma[m]}{\sum_{m'} \sigma[m']}, \qquad (2)$$

The notation $[m,:]$ indicates selecting all elements along the sequence dimension for the $m$-th modality. The colon ":" signifies selection across the entire dimension. The function $\mathtt{Mean}(\cdot)$ returns the average of the given vector. The variable $\mathbf{s}$ represents the attention weight for each modality. We define the number of uninformative tokens for the $m$-th modality as $K_m = \mathtt{floor}(\mathbf{s}[m] \times N)$. Thus, we prune $K_m$ tokens in the $m$-th modality and impute them using inter-modal tokens $\{\mathbf{x}_{m'}\}_{m' \neq m}^M$. This approach avoids setting a threshold for token fusion as in vanilla token fusion.

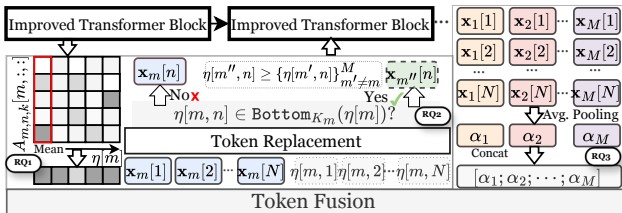

*Figure 3.* **Customized Token Fusion. RQ1**, **RQ2**, and **RQ3** correspond to the three research questions detailed in Section 4.3. We first identify the token score (Equation 3) and modality importance (Equation 2) via the attention distribution. Then we utilize this information to decide which token should be replaced and selected to impute intra-modal tokens (Equation 4). Finally, we use $\{x_m\}_{m=1}^M$ for the final prediction.

### 4.3. Customized Token Fusion

After obtaining the number of uninformative tokens for each modality, the next step is to perform token fusion. However, as discussed in Section 3, several challenges hinder the application of token fusion from visual fusion scenarios to broader medical applications. To adapt to medical multimodal learning, we propose a novel token fusion method, illustrated in Figure 3. In this subsection, we introduce our token fusion by addressing three specific research questions.

> **RQ1:** *How to identify uninformative tokens?*
> **A1:** *The attention distribution tells you.*

Unlike the original token fusion uses an additional scoring network with token-wise pruning loss to obtain the token score that indicate the informative level of tokens. Here, we use the attention distribution as the token score which avoids additional scoring network and corresponding token-wise

pruning loss. Specifically, the token score is obtained by:

$$\eta[m, n] = \max_{k}(A_{m,n,k}), \qquad (3)$$

where $\eta[m, n]$ denotes the token score for $m$-th modality, $n$-th token. This computing is overlapped with modality importance calculation, so we do not involve additional computation here. According to the token score $\eta[m, :]$, we select these $\texttt{Bottom-}K_m$ tokens as uninformative tokens.

> **RQ2:** *Which token from other modalities should replace an uninformative token?*
> **A2:** *Contrastive loss narrows the search space, then selecting the token with the highest score.*

Due to the heterogeneity between medical multi-modal data, the initial search space for the candidate replace token is $\{\mathbf{x}_{m'}\}_{m' \neq m}^{M}$ for the uninformative token $\mathbf{x}_m[n]$. The token in $\{\mathbf{x}_{m'}\}_{m' \neq m}^{M}$ contains both redundant and complementary. The token score $\eta$ cannot distinguish whether the token holds redundant or complementary information for $m$-th modality. Consequently, we introduce a two-level contrastive loss on the modality-specific encoded output. The goal of contrastive learning is to learn representations where positive inputs are closer in feature space than negative inputs. The first-level contrastive learning objective $\mathcal{L}_I$ uses features from the same instance as positive inputs and features from other instances as negative inputs (Li et al., 2022b). The second-level contrastive learning objective $\mathcal{L}_T$ uses the $n$-th token from different modalities of the same instance as positive inputs, while tokens from other instances serve as negative inputs (Zhou et al., 2024). The detailed definitions of these two-level contrastive learning objectives can be found in Appendix A.1.

The first-level contrastive learning improves the quality of representations. In the second-level contrastive learning, the goal is to encourage the $n$-th token across different modalities to capture the same information, eliminating the need for an explicit mapping relationship between modalities. This alignment ensures that tokens in $\{\mathbf{x}_{m'}[n]\}_{m' \neq m}^{M}$ can complement $\mathbf{x}_m[n]$. Consequently, the token replacement search space is narrowed from $\{\mathbf{x}_{m'}\}_{m' \neq m}^{M}$ to $\{\mathbf{x}_{m'}[n]\}_{m' \neq m}^{M}$. Additionally, the alignment in the second level reduces irrelevant information in $\{\mathbf{x}_{m'}[n]\}_{m' \neq m}^{M}$, mitigating information loss during the token replacement operation.

Note that the attention module is shared across all modalities, and input tokens are encouraged to align through contrastive learning objectives. As a result, we can compare attention values across different modalities, even if they are not processed in the same forward pass (multiple modalities are concatenated in the batch dimension, not the sequence dimension). This approach avoids randomly predefining which modality's token can replace the current token. Con-

sequently, the token fusion operation of $\mathbf{x}_m[n]$ is reformulated as follows:

$$
\begin{aligned}
\mathbf{x}_m[n] = \mathbf{x}_m[n] &\odot \mathbb{I}_{\eta[m,n] \notin \texttt{Bottom}_{K_m}(\eta[m])} \\
+ \sum_{m' \neq m}^{M} \mathbf{x}_{m'}[n] &\odot \mathbb{I}_{\eta[m,n] \in \texttt{Bottom}_{K_m}(\eta[m])} \odot \mathbb{I}_{\eta[m',n] \geq \eta\{m'',n\}_{m'' \neq m}^{M}}
\end{aligned} \quad (4)
$$

> **RQ3:** *How to use $\{\mathbf{x}_m\}_{m=1}^{M}$ for task prediction?*
> **A3:** *Average pooling and concatenate.*

After our transformer backbone, each $\mathbf{x}_m$ incorporates information from other modalities information via the self-attention module. Then, the challenge is whether to select one $\mathbf{x}_m$ from $\{\mathbf{x}_m\}_{m=1}^{M}$ or use the entire set for task prediction. In our $\texttt{AMC}$, we use the entire set $\{\mathbf{x}_m\}_{m=1}^{M}$ for the final prediction. Even with contrastive learning efforts to align modalities, modality-specific tokens can still carry heterogeneous information. The token replacement does not always ensure that the selected token $\mathbf{x}_{m'}[n]$ has the highest token score among $\mathbf{x}_{m'}$. Consequently, using only one $\mathbf{x}_m$ for task prediction might overlook critical and unique information from another modality $m'$. Specifically, we apply average pooling to each $\mathbf{x}_m$ to obtain $\mathbf{x}_m \in \mathbb{R}^D$. We then concatenate the set $\{\mathbf{x}_m\}_{m=1}^{M}$ to form a longer vector $\mathbf{z} \in \mathbb{R}^{DM}$. This concatenated vector $\mathbf{z}$ is subsequently fed into the task head for prediction.

### 4.4. Improved Transformer Block

Although our proposed $\texttt{AMC}$ provides the method to effectively remove uninformative tokens and retain critical information from important modalities for the final prediction, two challenges remain.

▷ *Diluted Attention Distribution.* Our method heavily relies on the attention distribution for modality importance calculation and token score assignment. However, the original token fusion can dilute the inner-modal attention weights of the self-attention mechanism (Wang et al., 2022b). This dilution can significantly undermine the final performance, potentially leading to inaccurate assessments of modality importance and token scores in $\texttt{AMC}$.

To address this problem, we employ the differential attention module (Ye et al., 2024). This approach is inspired by noise-canceling headphones and differential amplifiers in electrical engineering. The technique involves partitioning the query and key vectors into two groups, computing two separate attention maps, and then using the difference between these two maps as the final attention map. This differential operation can help reduce attention weight on irrelevant context. For simplicity, we illustrate this using single-head attention. Given the input sequence $\mathbf{x}_m$, the query, and key are calculated as follow:

$$[Q_1; Q_2] = \mathbf{x}_m W^Q, [K_1; K_2] = \mathbf{x}_m W^K, \qquad (5)$$

where $W^Q, W^K \in \mathbb{R}^{D \times 2D}$ are parameters, and $Q_1, Q_2, K_1, K_2 \in \mathbb{R}^{N \times D}$. Then the attention map calculation is re-formulated from $\texttt{softmax}(\frac{QK^T}{\sqrt{D}})$ to:

$$A = \texttt{softmax}(\frac{Q_1 K_1^T}{\sqrt{D}}) - \lambda \texttt{softmax}(\frac{Q_2 K_2^T}{\sqrt{D}}), \quad (6)$$

where the $\lambda$ is a learnable scalar. We use this attention map for our modality importance calculation and customized token fusion. For a detailed description of the differential transformer, including the multi-head extension, can refer to Appendix A.2.

▷ *Modality Conflict.* Intuitively, we can use modality-specific transformer layers for our token fusion, which is also the network architecture of the original token fusion paper. When the number of modalities increases, this design is parameter inefficient. However, using one branch of consecutive transformer layers to process all modalities introduces the gradient conflict optimization issue between modalities (Yun et al., 2024a;b). Therefore, we integrate the mixture-of-expert design to replace the original MLP layer in the transformer block to alleviate the gradient conflict optimization issue. For a detailed description of our mixture-of-experts design, please refer to Appendix A.3.

## 5. Experiment

**Setup.** To ensure a fair comparison with baselines, we use the best hyper-parameter settings from the original papers. If these are not available, we conduct hyper-parameter searches, including learning rate, hidden dimension, and batch size, with ranges of $[1e-3, 1e-4, 5e-5, 1e-5]$, $[32, 64, 128]$, and $[32, 64, 128]$, respectively. For our proposed method, we additionally search for the number of experts and the weights of $\mathcal{L}_I$, $\mathcal{L}_T$ and the load balancing loss of SMoE, with ranges of $[4, 8, 16]$, $[1.0, 0.1]$, $[1.0, 0.1]$, and $[1.0, 0.1]$, respectively. For the dataset split, we use $70\%$ for training, $15\%$ for validation, and $15\%$ for testing. All experiments were conducted using RTX 3090 GPUs. By default, each experiment was run five times with different seeds to ensure reproducibility, and the results were averaged. The final hyper-parameter settings for AMC are in Appendix B.1. For more implementation details for each dataset, see Appendix B.6.

### 5.1. Medical Application Evaluation

**Dataset.** We apply AMC on three medical multi-modal real-world datasets: Medical Information Mart for The Cancer Genome Atlas (TCGA), Intensive Care (MIMIC)-IV, and Alzheimer's Disease Neuroimaging Initiative (ADNI). We choose 5 common used cancer types in TCGA for the performance report.

*TCGA* involves data from 2,585 individuals across five cancer types: uterine corpus endometrial carcinoma (UCEC), lung adenocarcinoma (LUAD), brain lower grade glioma (LGG), breast invasive carcinoma (BRCA), and bladder urothelial carcinoma (BLCA). We use three modalities—whole slide images (WSIs), pathology reports, and RNA-seq data—to predict patient survival (time-to-event) separately for each cancer type. WSIs are histopathological images containing spatial and morphological information vital for diagnosis and prognosis. Pathology reports, which are text-based descriptions of tumor samples, are vectorized before inputting to our model. Gene expression profiles measured via bulk RNA-seq provide molecular-level insights into pathophysiology. We do the patient survival time prediction on this dataset and report the c-index metric (higher is better) (Alabdallah et al., 2024).

*MIMIC-IV* is a large database of de-identified patient data from the emergency department or intensive care unit at the Beth Israel Deaconess Medical Center in Boston, MA. Our dataset consists of 9,003 patients with laboratory measurements (denoted as Lab), clinical notes (denoted as Note), and billing codes (denoted as Code). Using these modalities, we predict patient mortality within one year of admission, framing it as a binary classification problem.

*ADNI* includes biofluid biomarker measurements, clinical assessments, genomic data, and neuroimages from patients diagnosed as cognitively normal (CN), with mild cognitive impairment (MCI), or Alzheimer's dementia (AD). Biospecimen data include biomarkers from plasma, serum, cerebral spinal fluid (CSF), and urine samples. Clinical assessments consist of cognitive tests, questionnaires, physical exam measurements, and medical history. Genomic data contain genotype dosages for 135,595 genetic variants. Neuroimages are structural MP-RAGE MRIs aligned to an average template and segmented using MUSE. Our dataset consists of 2,380 subjects, and the inference task is diagnosis classification, framed as a three-class problem.

**Result.** TCGA (Table 1): AMC attains the highest c-index on three cancer types {UCEC, LUAD, BLCA} with {$0.068 - 0.510, 0.009 - 0.182, 0.025 - 0.128$} improvement, ranks the third best performance on LGG cancer, and the second best performance on the BRCA cancer. Overall, AMC achieves the best average C-index across all cancer types. These results demonstrate that our method consistently outperforms existing multi-modal fusion baselines in survival prediction accuracy. ADNI (Table 2): We achieve improvements of {$3.64 - 16.13, 3.32 - 14.3, 2.23 - 22.74$} in terms of Accuracy, Recall, and F1 metrics. These improvements show that our method is reliable for the majority class and also effectively identifies rare but important cases, making it useful for general applications. However, our method scores lower than MUSE and FlexMOE in Precision, suggesting that more attention is needed to reduce false posi-

*Table 1.* Performance on TCGA dataset. The **bold** denotes the best performance and the underline denotes the second-best performance.

| Cancer Type | FlexMOE | FuseMOE | LiMOE | MAGGate | MulT | TF | MUSE | Proposed |
|---|---|---|---|---|---|---|---|---|
| UCEC | 0.510±0.043 | 0.353±0.117 | 0.795±0.050 | 0.613±0.223 | 0.665±0.071 | 0.722±0.051 | 0.501±0.46 | **0.863±0.023** |
| LUAD | 0.502±0.024 | 0.615±0.047 | 0.674±0.024 | 0.621±0.031 | 0.616±0.017 | 0.664±0.026 | 0.501±0.25 | **0.683±0.030** |
| LGG | 0.504±0.056 | 0.567±0.08 | 0.771±0.023 | 0.799±0.019 | **0.828±0.019** | 0.696±0.031 | 0.499±0.22 | 0.795±0.019 |
| BRCA | 0.464±0.066 | 0.49±0.09 | **0.736±0.018** | 0.684±0.085 | 0.725±0.008 | 0.721±0.018 | 0.501±0.26 | 0.727±0.009 |
| BLCA | 0.478±0.086 | 0.535±0.068 | 0.578±0.055 | 0.475±0.068 | 0.569±0.037 | 0.565±0.075 | 0.501±0.00 | **0.603±0.055** |
| Average | 0.492 | 0.512 | 0.711 | 0.638 | 0.681 | 0.674 | 0.501 | **0.734** |

*Table 2.* Performance on ADNI dataset.

| | Acc | Recall | Precision | F1 |
|---|---|---|---|---|
| FlexMOE | 52.27±2.99 | 48.97±6.41 | 57.58±2.67 | 47.47±9.56 |
| FuseMOE | 48.85±4.18 | 49.81±3.51 | 52.76±2.05 | 49.65±3.88 |
| LiMOE | 46.55±7.56 | 41.69±4.10 | 51.92±6.44 | 34.80±8.20 |
| MAGGate | 47.68±7.71 | 46.48±7.71 | 50.99±4.93 | 40.79±11.80 |
| MulT | 44.87±7.85 | 40.73±6.08 | 51.00±4.34 | 32.08±10.42 |
| TF | 39.78±9.08 | 41.85±4.87 | 45.22±3.67 | 35.27±10.20 |
| MUSE | 51.51±1.25 | 51.71±1.82 | **58.72±0.91** | 52.59±1.65 |
| Proposed | **55.91±2.49** | **55.03±2.58** | 55.38±2.55 | **54.82±2.41** |

*Table 3.* Performance on MIMIC-IV dataset.

| | Acc | Recall | Precision | F1 |
|---|---|---|---|---|
| FlexMOE | 74.57±0.74 | 51.76±1.21 | 54.07±2.01 | 49.96±2.31 |
| FuseMOE | 77.31±0.22 | 50.20±0.22 | 53.08±14.11 | 44.35±0.61 |
| LiMOE | 69.99±2.24 | 56.88±1.33 | 57.13±1.02 | 56.69±1.04 |
| MAGGate | 77.42±0.0 | 50.00±0.0 | 38.71±0.0 | 43.64±0.0 |
| MulT | 68.66±1.15 | 58.62±0.57 | 57.69±0.74 | 57.96±0.72 |
| TF | **77.41±0.03** | 49.99±0.02 | 38.71±0.00 | 43.63±0.01 |
| MUSE | 66.70±0.74 | 50.02±1.87 | 33.93±0.42 | 40.38±0.38 |
| Proposed | 68.51±1.59 | **61.65±0.99** | **59.46±0.81** | **59.75±1.00** |

*Table 4.* Different modality importance calculation methods.

| | Acc | Recall | Precision | F1 |
|---|---|---|---|---|
| `AMC` | 68.51±1.59 | 61.65±0.99 | 59.46±0.81 | **59.75±1.00** |
| `AMC w/ STD` | 67.24±2.00 | **62.24±1.05** | **59.50±1.21** | 59.53±1.21 |
| `AMC w/ Max` | **70.23±3.65** | 59.46±4.77 | 55.40±8.35 | 56.66±6.52 |

*Table 5.* Different token score assignment methods.

| | Acc | Recall | Precision | F1 |
|---|---|---|---|---|
| `AMC` | **68.51±1.59** | **61.65±0.99** | **59.46±0.81** | **59.75±1.00** |
| `AMC` w/o Differential Attention | 65.17±1.88 | 61.54±0.80 | 58.59±0.55 | 58.03±1.01 |
| `AMC` w/ Original Token Fusion | 66.41±1.21 | 61.59±1.58 | 58.89±0.49 | 58.82±0.74 |

## 5.2. In-depth Analysis

In this subsection, we conduct in-depth analysis experiments to show the insights provided by `AMC`. Specifically, we evaluate the effect of different methods for assigning modality importance (Paragraph 1), and token score (Paragraph 2). Then, we provide the token exchange (Paragraph 3) and the modality score (Paragraph 4) analysis.

**1. Modality Importance Calculation.** To determine modality importance, a key operation is an average (`Mean`) to capture the statistical information of attention distribution for each modality. Intuitively, the standard deviation (`STD`) could also represent this statistical information. Additionally, we can use `Max` to consider the maximum token score of the modality as its importance.

We compared the performance differences between `Mean`, `STD`, and `Max`. The results in Table 12 show that `Mean` and `STD` yield similar results. `Mean` performs better in Accuracy and F1 metrics, while `STD` is better in Recall and Precision. Overall, `Mean` is slightly better, with a higher F1 metric and lower standard deviation across all four metrics. The use of `Max` is clearly worse than both `Mean` and `STD`, as it results in lower Recall, Precision, and F1 metrics. This indicates that statistical information about token scores better reflects how each modality contributes to the final prediction.

**2. Token Score Assignment.** Here, we compare our method with the original token fusion method (the "`AMC` w/ Original Token Fusion" in Table 5) in the token score assignment. The original token fusion method uses an individual linear network to predict the token score, while we use

tives. MIMIC-IV (Table 3): We observe improvements of $\{3.03 - 11.66, 1.77 - 25.53, 1.79 - 19.37\}$ in Recall, Precision, and F1, respectively. These gains demonstrate that our method achieves more reliable and precise predictions for the target class, particularly in scenarios where minimizing false negatives and false positives is critical. The slightly lower overall accuracy compared to baselines suggests that prior methods may be overly reliant on the majority class, potentially reflecting overfitting to class-imbalanced training data rather than generalizing to meaningful patterns.

▷ *Non-Medical Task Validation.* We apply our method to a traditional multi-modal dataset to test the generalizability of the proposed `AMC` and network design. Due to space constraints, the results are provided in Appendix B.2. These results demonstrate that our method remains effective in non-medical tasks.

the attention map from the previous transformer layer for this purpose. Our approach employs a differential attention module to enhance attention map quality and mitigate the issue of over-allocating attention to irrelevant contexts. We also consider using a vanilla attention module to obtain the attention map for token score assignment as an additional method (the "AMC w/o Differential Attention" in Table 5).

The results in Table 5 show that using the attention map from differential attention to calculate the token score is better than using an individual linear layer (AMC outperforms "AMC w/ Original Token Fusion" in Accuracy, Recall, Precision, and F1 metrics). The vanilla attention map faces the over-allocation problem, resulting in poorer performance ("AMC w/ Original Token Fusion" is better than "AMC w/o Differential Attention" in all metrics). This indicates that the over-allocation issue decreases the quality of using attention maps for token score calculation.

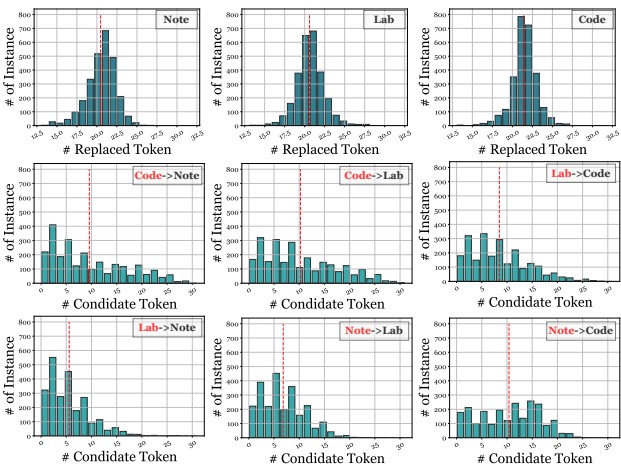

*Figure 4.* **The statistic of token replacement on the MIMIC-IV testing set.** The first row shows the distribution of tokens in each modality that have been selected for replacement by tokens from other modalities. The last two rows show the distribution of the number of tokens where a modality was chosen to replace another modality's tokens. Since token fusion is applied to each modality's tokens, denoted as $\mathbf{a}_m$, and MIMIC-IV includes three modalities in total, we have two rows for this data. In the format "M1→M2"," M1" indicates the candidate modality, while "M2" represents the modality that will be replaced by modality "M1". The dashed red vertical line indicates the average value.

**3. Token Exchange Analysis** We visualize the token exchange statistics in Figure 4. The results show that more tokens from the billing codes modality (referred to as Code) are replaced by inter-modal tokens. It shows that the number of tokens replaced varies across modalities, with Code having a higher frequency of token replacement. The last two rows of Figure 4 demonstrate the dynamics of token replacement, where one modality replaces another. For example, tokens from the Code modality frequently replace

tokens in the Note modality, as seen in the "Code→Note" plot. The detailed distribution of token replacements and contributions makes the fusion process more transparent. This transparency can be communicated to stakeholders, such as clinicians, to build trust in automated systems that leverage multi-modal data for decision-making.

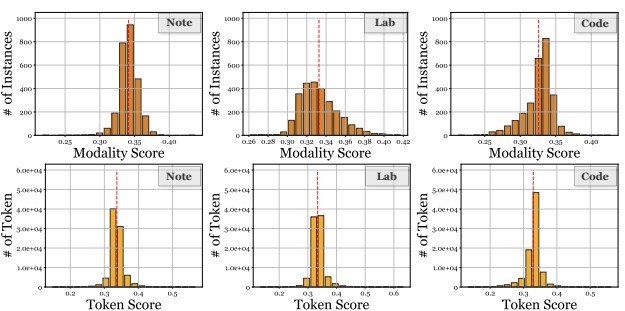

*Figure 5.* **The modality importance (Top) and token score (Bottom) statistics.** The red dash line is the average value of modality score and token score.

**4. Modality Importance Visualization.** In Figure 5, the top row shows the distribution of modality scores for Note, Lab, and Code. Each modality has a distinct score distribution. This indicates they contribute differently to the model's decisions. For example, the Code modality has more instances with higher scores. This suggests it often provides more valuable information. In contrast, the Lab modality has a broader and flatter distribution. The bottom row presents the distribution of token scores for each modality. Note and Code have narrower distributions around their means compared to Lab. This implies more consistent token quality in these modalities. The Lab modality shows more variation in token informativeness. This variation can affect the overall contribution of the Lab modality. Visualizing modality importance and token scores is useful for building interpretable and efficient models. It is especially important in applications where understanding the information source impacts decision-making.

### 5.3. Ablation Study

We conducted an ablation study to examine the impact of the four key modules proposed by AMC on performance. The results in Table 6 indicate that removing Contrastive Loss and Differential Attention leads to lower F1 scores. This highlights their effectiveness in enhancing token fusion. The decrease in performance when SMoE is removed underscores its role in managing gradient conflicts between modalities. This results in the lowest F1 and Precision metrics. Lastly, the drop in performance when Token Fusion is excluded demonstrates its importance in balancing contributions from different modalities.

*Table 6.* The ablation study on the MIMIC-IV dataset.

| Token Fusion | Contrasive Loss | Differential Attention | SMoE | Acc | Recall | Precision | F1 |
|---|---|---|---|---|---|---|---|
| √ | √ | √ | √ | **68.51±1.59** | 61.65±0.99 | **59.46±0.81** | **59.75±1.00** |
|  | √ | √ | √ | 66.97±1.67 | **62.94±0.37** | 59.24±0.23 | 57.71±0.90 |
| √ |  | √ | √ | 64.90±0.86 | 62.24±0.85 | 59.00±0.49 | 58.44±0.37 |
| √ | √ |  | √ | 65.17±1.88 | 61.54±0.80 | 58.59±0.55 | 58.03±1.01 |
| √ | √ | √ |  | 65.27±2.37 | 60.78±1.12 | 58.23±0.43 | 57.71±0.79 |

## 6. Conclusion

We propose `AMC`, a dynamic fusion method that first identifies the importance of each modality and then fuses them accordingly. Our experiments demonstrate that `AMC`, along with the corresponding network architecture, can effectively fuse multi-modal information in the medical domain. Additionally, our method provides interpretable insights into how each modality and token contributes to the final prediction.

## Impact Statement

This paper aims to advance the field of Medical Machine Learning. All datasets used are publicly available. While our work may have societal consequences, none need to be specifically highlighted here.

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

# A. Method Details

## A.1. Definition of Two-level Contrastive Learning objectives

Given $B$ pairs data from $m$-th and $m'$-th modality, the corresponding tokens is $\mathcal{Z}_B = \{\mathbf{a}_m^i, \mathbf{a}_{m'}^i\}_{i=1}^B$, where $\mathbf{a}_m^i \in \mathbb{R}^{N \times D}$. We denote the instance-level features as the average pooling of $\mathbf{a}_m^i$ across the first dimension for each elements, resulting in $\mathbf{x}_m^i \in \mathbb{R}^D$. Then we define the instance-level pair-wised representation as $\mathcal{X}_B = \{\mathbf{x}_m^i, \mathbf{x}_{m'}^i\}_{i=1}^B$. The contrastive training objective at the instance level with pre-defined temperature $T$ is:

$$\mathcal{L}_i(\mathcal{X}_B) = -\frac{1}{2} \log \frac{e^{\langle \mathbf{x}_m^i, \mathbf{x}_{m'}^i \rangle / T}}{\sum_{j=1}^B e^{\langle \mathbf{x}_m^i, \mathbf{x}_{m'}^j \rangle / T}} - \frac{1}{2} \log \frac{e^{\langle \mathbf{x}_m^i, \mathbf{x}_{m'}^i \rangle / T}}{\sum_{j=1}^B e^{\langle \mathbf{x}_m^j, \mathbf{x}_{m'}^i \rangle / T}}, \tag{7}$$

where the $\langle \cdot, \cdot \rangle$ denotes the dot product operation. Then contrastive leartning objective at the token level is temperature $T$ is:

$$\mathcal{L}_{i,n}(\mathcal{Z}_B) = -\frac{1}{2} \log \frac{e^{\langle \mathbf{a}_m^i[n], \mathbf{a}_{m'}^i[n] \rangle / T}}{\sum_{j=1}^B \sum_{n'}^N e^{\langle \mathbf{a}_m^i[n], \mathbf{a}_{m'}^j[n'] \rangle / T}} - \frac{1}{2} \log \frac{e^{\langle \mathbf{a}_m^i[n], \mathbf{a}_{m'}^i[n] \rangle / T}}{\sum_{j=1}^B \sum_{n'}^N e^{\langle \mathbf{a}_m^j[n'], \mathbf{a}_{m'}^i[n] \rangle / T}}, \tag{8}$$

where the $\mathbf{a}_m^i[n]$ denotes the $n$-th token of $\mathbf{a}_m^i$. Consequently, the $\mathcal{L}_I$ and $\mathcal{L}_T$ is defined as:

$$\mathcal{L}_I = \sum_{i=1}^B \mathcal{L}_i(\mathcal{X}_B), \mathcal{L}_T = \sum_{i=1}^N \sum_{n=1}^N \mathcal{L}_{i,n}(\mathcal{Z}_B). \tag{9}$$

## A.2. Differential Self-Attention Details

Given the input sequence $\mathbf{a}_m$, the query, and key is calculated via:

$$[Q_1; Q_2] = \mathbf{a}_m W^Q, [K_1; K_2] = \mathbf{a}_m W^K, V = \mathbf{a}_m W^V \tag{10}$$

where $W^Q, W^K, W^V \in \mathbb{R}^{D \times 2D}$ are parameters, $Q_1, Q_2, K_1, K_2 \in \mathbb{R}^{N \times D}, V \in \mathbb{R}^{N \times 2D}$. Then the attention map calculation is re-formulated from $\texttt{softmax}(\frac{QK^T}{\sqrt{D}})$ to:

$$A = \texttt{softmax}(\frac{Q_1 K_1^T}{\sqrt{D}}) - \lambda \texttt{softmax}(\frac{Q_2 K_2^T}{\sqrt{D}}). \tag{11}$$

Then the attention module output is: $\texttt{DAttn}(\mathbf{a}_m) = AV$. The $\lambda$ is defined as:

$$\lambda = \exp(\lambda_{q_1} \cdot \lambda_{k_1}) - \exp(\lambda_{q_2} \cdot \lambda_{k_2}) + \lambda_{init}, \tag{12}$$

where $\lambda_{q_1}, \lambda_{q_2}, \lambda_{k_1}, \lambda_{k_2} \in \mathbb{R}^D$ are learnable vectors, and $\lambda_{init} \in (0, 1)$ is a constant number. *We set $\lambda_{init} = 0.1$ in default.*

**Multi-Head Extension.**  When we have $h$ attention heads, the projection parameters is defined as $W_i^Q, W_i^K, W_i^V, i = \{0, 1, \cdots, h\}$. The scalar $\lambda$ is shared cross heads within the same layer. The computation pipeline results as follows:

$$[Q_1^i; Q_2^i] = \mathbf{a}_m W_i^Q, [K_1^i; K_2^i] = \mathbf{a}_m W_i^K, V^i = \mathbf{a}_m W_i^V,$$
$$A_i = \texttt{softmax}(\frac{Q_1^i (K_1^i)^T}{\sqrt{D}} - \lambda \texttt{softmax}(\frac{Q_2^i (K_2^i)^T}{\sqrt{D}}),$$
$$\texttt{head}_i = A_i V_i$$
$$\overline{\texttt{head}}_i = (1 - \lambda_{init}) \cdot \texttt{LN}(\texttt{head}_i),$$
$$\texttt{MultiHead}(\mathbf{a}_m) = \texttt{Concat}(\overline{\texttt{head}}_1, \cdots, \overline{\texttt{head}}_h) W^O,$$

where $W^O \in \mathbb{R}^{D \times D}$ is the learnable projection layer, $\texttt{LN}(\cdot)$ uses RMSNorm (Zhang & Sennrich, 2019) for each head, and $\texttt{Concant}(\cdot)$ concatenates the heads together along the channel dimension. We compute the average of $\{A_1, \cdots, A_h\}$ to obtain the attention mapp $A$, which is used to determine modality importance and perform customized token fusion.

### A.3. The Mixture-of-Expert in the Transformer Block

In our transformer block, we replace the MLP layer with a Mixture-of-Experts (MoE) module, consisting of multiple experts denoted as $\{f_1, f_2, \cdots, f_E\}$, where $E$ is the number of experts. We employ the sparse activated MoE (SMoE) variant, which includes a router $\mathcal{R}$. The router is responsible for the routing mechanism, sparsely selecting the top-$k$ experts. For a given token $\mathbf{x}$, the router $\mathcal{R}$ selects the top-$k$ experts based on scores derived from a softmax function applied to a learnable gating function $g(\cdot)$. We implement this gating function using a single linear layer. The router then outputs $\mathcal{R}(\mathbf{x})$, , identifying the indices of the selected experts. This process is described as follows::

$$\mathbf{y} = \sum_{i=1}^{E} \mathcal{R}(\mathbf{x})_i \cdot f_i(\mathbf{x}), \tag{13}$$

$$\mathbf{x} = \text{Top-K}(\text{softmax}(g(\mathbf{x})), k), \tag{14}$$

$$\text{Top-K} = \begin{cases} \mathbf{v}, \textit{if } \mathbf{v} \textit{ is in the top } k, \\ 0, \textit{otherwise}. \end{cases} \tag{15}$$

where $g(\mathbf{x})$ computes the scores for each expert, and the top-$k$ function selects the experts with the highest scores. To avoid the load imbalance problem in SMoE, we use the load and imbalance loss (Shazeer et al., 2022).

## B. Experiment Details

### B.1. Hyper-parameter on Each Dataset

Table 7. The hyper-parameter setup for AMC.

| | ADNI | MIMIC-IV | TCGA | | | | | ENRICO |
| | | | UCEC | LUAD | LGG | BRCA | BLCA | |
| --- | --- | --- | --- | --- | --- | --- | --- | --- |
| Learning rate | 1e-4 | 1e-3 | 1e-3 | 1e-3 | 1e-3 | 1e-3 | 1e-3 | 5e-3 |
| # of Experts | 8 | 8 | 8 | 8 | 8 | 8 | 8 | 8 |
| Top-K | 2 | 2 | 2 | 2 | 2 | 2 | 2 | 2 |
| # of Transformer Layers | 2 | 2 | 2 | 2 | 2 | 2 | 2 | 4 |
| Training Epochs | 30 | 100 | 30 | 30 | 30 | 30 | 30 | 100 |
| Warm-up Epochs | 5 | 10 | 5 | 5 | 5 | 5 | 5 | 5 |
| Hidden dimension | 64 | 64 | 128 | 64 | 64 | 64 | 64 | 128 |
| Batch Size | 32 | 64 | 64 | 64 | 64 | 64 | 64 | 128 |
| # of Attention Heads | 8 | 8 | 8 | 8 | 8 | 8 | 8 | 8 |

### B.2. Non-Medical Multi-modal Evaluation

**Dataset.** We select Enhanced Rico (ENRICO) dataset (Leiva et al., 2021) evaluates the generalizability of AMC. The ENRICO is the dataset of Android app screens categorized by their design motifs. It contains $1,460$ Android app screens (image modality) and the corresponding view hierarchy (set modality). The instances in this ENRICO can be categorized into 20 design categories.

**Result.** We compare existing fusion methods tested on this dataset as described in (Liang et al., 2021; Paul Pu Liang et al., 2022). We use the same evaluation setup as in these references, which includes searching for hyper-parameters like learning rate and hidden dimension. We report the mean and standard deviation from 10 repetitions, ensuring a fair comparison with the results from (Liang et al., 2021; Paul Pu Liang et al., 2022). All baseline results are taken from the original papers (Liang et al., 2021; Paul Pu Liang et al., 2022). However, we lack the standard deviation for HighMMT because it was not reported in the original paper. From Table 8, we see that our method, AMC, shows an improvement of $\{16.3 - 24.5\}$ over multi-modal baselines and $\{21.9 - 22.8\}$ over single-modal baselines. This result demonstrates that our method is also effective in a non-medical multi-modal benchmark.

*Table 8.* Performance on the ENRICO dataset. We refer to the evaluation metric on previous works (Liang et al., 2021; Paul Pu Liang et al., 2022), and report the 20 classes' prediction accuracy (%).

| | Unimodal (MultiBench) | Unimodal (MultiBench) | Late Fusion | TensorFusion | LRTF | MI-Matrix | CCA | RefNet | GradBlend | HighMMT | Proposed |
|---|---|---|---|---|---|---|---|---|---|---|---|
| Modality | Image | Set | Image, Set | | | | | | | | |
| Acc (20) | 47.0±1.6 | 46.1±1.3 | 50.8±2.0 | 46.6±1.9 | 47.1±2.9 | 46.7±2.4 | 50.1±1.4 | 44.4±2.2 | 51.0±1.4 | 52.6 | 68.9±3.7 |

*Table 9.* Extend AMC for the modality missing scenario.

| | Acc | Recall | Precision | F1 |
|---|---|---|---|---|
| AMC | 55.91±2.49 | 55.03±2.58 | 55.38±2.55 | 54.82±2.41 |
| AMC w/ missing modality | 56.3±2.21 | 55.77±3.32 | 57.39±2.29 | 54.81±2.18 |

## B.3. Extra Experiments

**Modality Missing Scenario.**    We extend AMC to solve the missing modality problem. In such cases, we treat the importance score and token score of any missing modality as zero. This approach enables AMC to function effectively even when some data is unavailable. To demonstrate AMC's adaptability to the missing modality scenario, we conducted experiments using the ADNI dataset. This dataset is naturally suited for testing missing modality scenarios, providing a robust environment to validate our approach. The results in the Table 9 indicates that AMC's performance decreases only slightly when dealing with missing modalities. This demonstrates that AMC can effectively handle the missing modality problem through simple extensions. We use the same model but add examples with missing modality problems. The similar performance indicates that AMC still keeps its prediction capability in these additional examples.

**Efficiency Experiment.**    We report the Mean time per iteration during training, and GFLOPs during testing across baselines and AMC. The result in Table 10 shows our method is computationally efficient.

**Modality Importance Varies Across Tasks.**    We evaluate the single modality task performance across MIMIC-IV and ADNI. The results in Table 12 show that the importance of different modalities varies across different tasks.

## B.4. Importance of Modality Score

**Dataset Level Modality Importance.**    We assess modality importance at the dataset level using the MIMIC-IV dataset. Each modality is evaluated individually using the same AMC backbone. The results in Table 12 indicates that the 'Code' modality is less important (the lowest F1, Recall, and Precision), aligning with Figure 4 where 'Code' has a higher frequency of token replacement.

**Instance Level Modality Measurements.**    We investigate the effectiveness of our modality score in reducing prediction uncertainty. We compare the uncertainty of predictions with their original modality scores against AMC with equal modality scores. The uncertainty of predictions is defined as $-\sum_{i=1}^{N} p_i \log(p_i)$, where $p_i$ is the predicted probability of class $i$. Our findings show that the modality importance score effectively reduces prediction uncertainty (from 0.1218 increase to 0.1343), demonstrating its utility in identifying significant modalities.

## B.5. Interpretability Experiment

**Gradient Conflict.**    We analyze the distribution of cosine distances between training gradients derived from the different modalities in the MIMIC-IV dataset. These gradients are computed from a model configured with experts and a dense MLP, both following the same configuration as in the AMC and Dense Model setups. The gradients are extracted from the final transformer layer. Higher positive cosine distances indicate reduced gradient conflict, which suggests better alignment between the modalities in terms of learning direction. Our result in Figure 6 shows that the SMoE effectively alleviates the gradient conflict between different modalities.

**Modality Importance (Case Study).**    We propose a case study using images from the TCGA dataset. The image is easier for readers to realize the modality data quality. This study illustrates the variation in modality importance, providing a visual representation of data quality. The results in Figure 7 show that images with lower modality importance scores generally

*Table 10.* The computation efficiency comparison.

| Metric | FlexMOE | FuseMOE | LiMOE | MAGGate | MulT | TF | AMC |
|---|---|---|---|---|---|---|---|
| Mean Time (s) | 12.73 | 18.68 | 12.65 | 11.64 | 12.85 | 12.4 | **11.5** |
| GFLOPs | 59.07 | 59.76 | 59.41 | 59.06 | 60.12 | 59.39 | **45.23** |

*Table 11.* The modality importance across different tasks.

| MIMIC-IV | Code | Note | Lab | ADNI | Genomic | Image | Biospecimen | Clinical |
|---|---|---|---|---|---|---|---|---|
| Acc | **67.95** | 67.58 | 64.77 | Acc | 52.94 | **53.21** | 50.4 | **53.50** |
| Precision | 55.44 | 55.82 | **59.18** | Precision | 53.23 | **55.91** | 49.1 | 53.30 |
| Recall | 55.88 | 56.51 | **62.50** | Recall | 53.55 | **54.70** | 32.6 | 52.43 |
| F1 | 55.60 | 56.02 | **58.67** | F1 | 52.30 | **54.12** | 39.1 | 47.32 |

exhibit poor quality, while images with higher scores indicate better quality.

**Token Replacement (Case Study).** We provide several case studies about how each modality's tokens are replaced by other modalities' tokens in the Figure 8.

### B.6. Implementation Details

The implementation difference between each dataset is the modality-specific encoder.

**MIMIC-IV.** In MIMIC-IV, each modality is represented as a feature vector. We begin by splitting each feature vector into $N$ sub-vectors and projecting these sub-vectors into the model's hidden dimension. Learnable positional embeddings (Wang et al., 2022a) are then added to transform these projections into input tokens. To enhance feature quality, we first pass these tokens through modality-specific vanilla transformer layers. The outputs from these layers are then fed into our improved transformer blocks, where token fusion is performed to integrate the modalities effectively. Finally, these integrated tokens are used to make the final prediction.

**ADNI.** In ADNI, genomic, biospecimen, and clinical modalities are represented as feature vectors. As with MIMIC-IV, we split each feature vector into $N$ sub-vectors which are projected into the model's hidden dimension. The imaging modality is represented as a 3D tensor to which we first apply multiple 3D convolutions with batch normalization. This is followed by average pooling and a linear layer projection to produce a feature vector that is split into $N$ sub-vectors in the model's hidden dimension. The resulting representations are in the same format as the other modalities and undergo the same downstream steps as described for MIMIC-IV. Namely, learnable positional embeddings are added to the vectors to produce input tokens, which are then passed through modality-specific vanilla transformer layers. This is followed by token fusion via our improved transformer blocks, resulting in integrated tokens that are utilized for prediction.

**TCGA.** In TCGA, each modality is represented as a feature vector. RNA-seq data is already tabular, whereas the WSIs and text data were preprocessed to extract feature vectors. Inputs are first projected into the model's hidden dimension and then split into $N$ sub-vectors. Query tokens are learned by applying a single transformer layer with attention between these tokens and the sub-vectors. The resulting outputs are $N$ tokens per modality. The following steps are the same as outlined above: learnable positional embeddings are added to these input tokens, these are then input to modality-specific vanilla

*Table 12.* Dataset-level modality importance on the MIMIC-IV dataset.

| Modality | Code | Note | Lab |
|---|---|---|---|
| Acc | 67.95 | 67.58 | 64.77 |
| Precision | 55.44 | 55.82 | 59.18 |
| Recall | 55.88 | 56.51 | 62.50 |
| F1 | 55.60 | 56.02 | 58.67 |

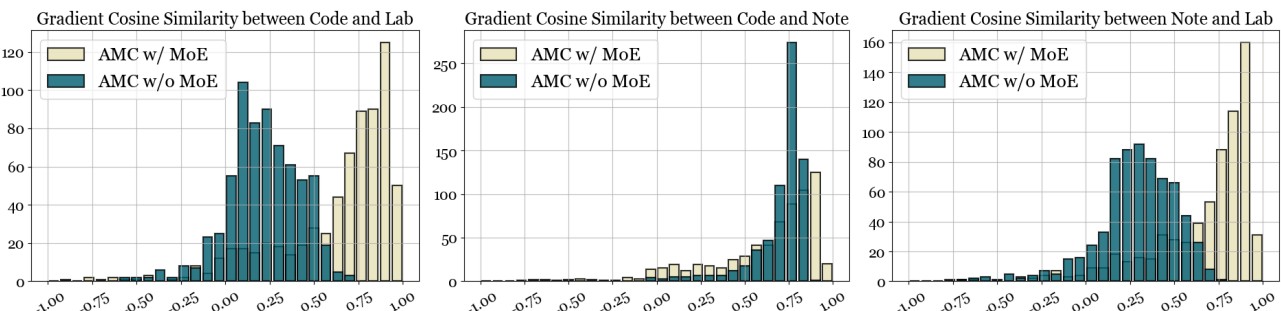

*Figure 6.* **Gradient Conflict.** The gradient conflict between different modalities in the MIMIC-IV dataset. The "AMC w/ MoE" is the proposed `AMC` method, and "AMC w/o MoE" replaces the MoE to its dense counterpart.

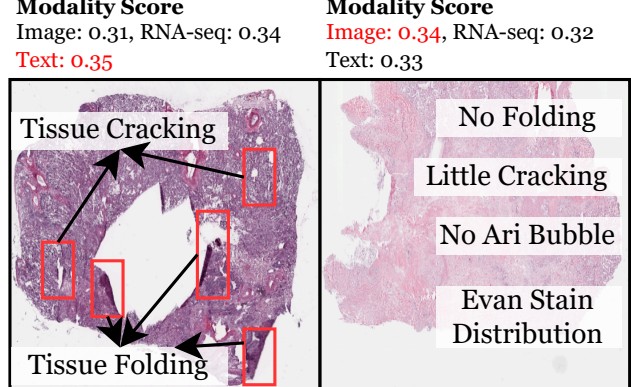

*Figure 7.* **Modality Importance.** The case study about the modality data quality identified by our proposed modality importance. The image on the left side obtains the lowest modality importance, and the right side image obtains the highest modality importance.

transformer layers, and token fusion is applied to generate integrated tokens for the prediction task.

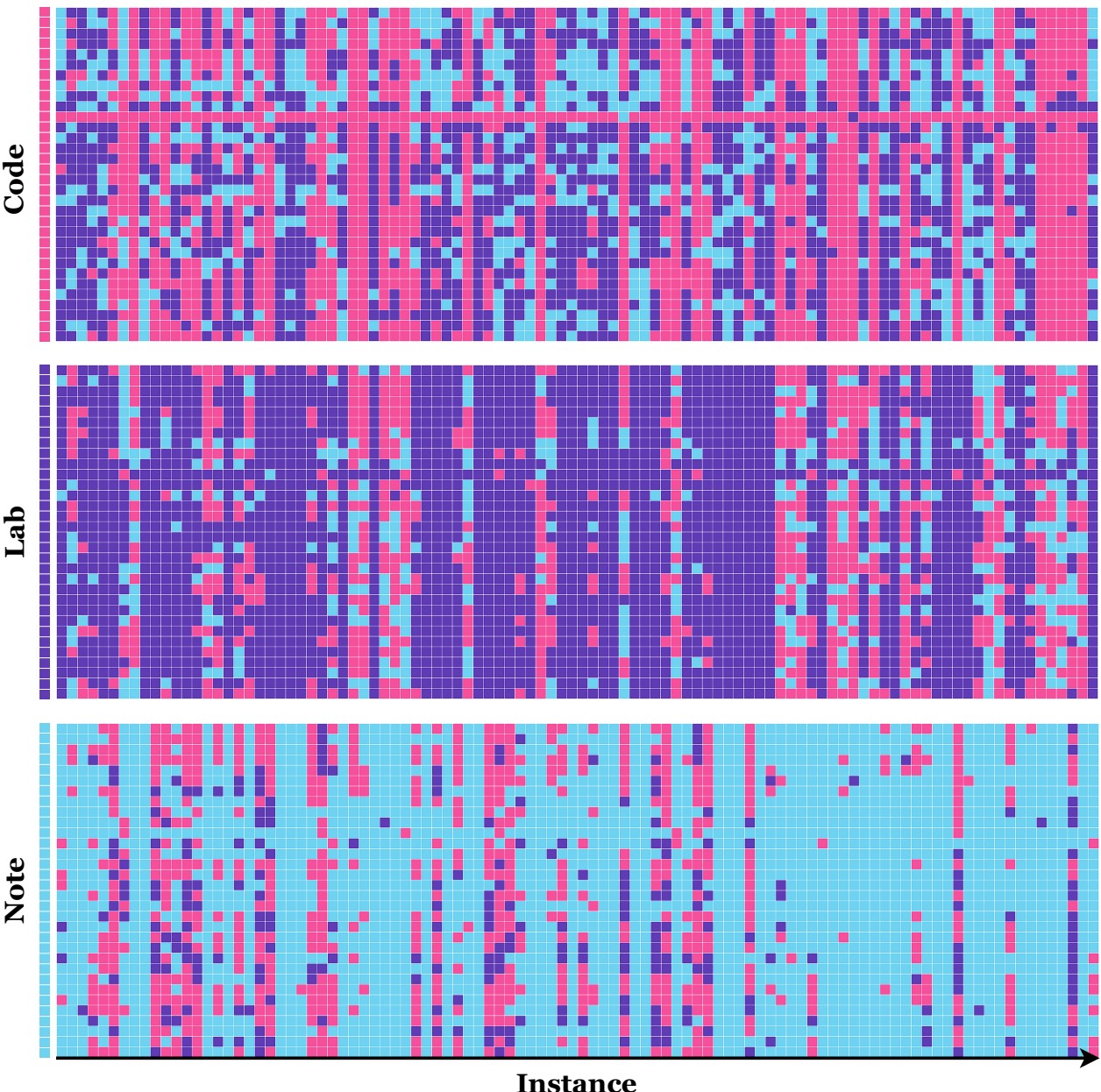

*Figure 8.* **Token Replacement.** The case study on token replacement within the MIMIC-IV dataset. Tokens are color-coded by modality: red blocks represent Code tokens, purple blocks indicate Lab tokens, and blue blocks signify Note tokens. The first row shows the token sequence before replacement. Each subsequent row displays the token sequence after replacement for a different instance. We randomly select 100 instances from the testing set for the visualization.

