# OpenReview forum: "Modalities Contribute Unequally: Enhancing Medical Multi-modal Learning through Adaptive Modality Token Re-balancing"
_ICML.cc/2025/Conference — ICML 2025 poster_

### Official Review · Reviewer_CN7m · 2025-03-09

**Overall Recommendation:** 3

**Summary:**

This paper proposed AMC, a top-down dynamic multi-modal fusion approach. AMC firstly quantifies the importance of each modality and then fuse the information from different modalities based on the importance. Experimental results show that AMC is effective on several public datasets. This paper addresses a significant challenge in the field of medical multi-modal learning: effectively fusing heterogeneous data from various modalities, especially under conditions where their data quality can vary.

**Claims And Evidence:**

The paper makes several claims regarding the effectiveness of the Adaptive Modality Token Re-Balancing (AMC) method in enhancing medical multi-modal learning. Most of the claims are supported well.

However, 'its interpretability in AI4Medical field' requires supplementation with specific clinical interpretability evaluation.

Besides, claims regarding computational efficiency and scalability are supported by theoretical explanations but would benefit from empirical validation, especially in terms of runtime and resource usage on large-scale datasets.

**Essential References Not Discussed:**

No.

**Experimental Designs Or Analyses:**

Experimental designs are valid.

**Methods And Evaluation Criteria:**

The proposed methods and/or evaluation criteria make sense for the problem. The choice of datasets and evaluation metrics provides a comprehensive understanding of the AMC framework’s capabilities and limitations.

**Other Comments Or Suggestions:**

N/A

**Other Strengths And Weaknesses:**

Strengths: Proposes a top-down dynamic fusion method combining modality-level weighting with token-level replacement. The results shows this method is effective on several public datasets.
Weaknesses: Limited novelty. While the paper integrates various existing techniques from multi-modal learning, attention mechanisms, and contrastive learning, the method seems to be a combination of different techniques. And there should be more valid proof or evaluation of the claim about interpretability.

**Questions For Authors:**

Can you explain more about your novelty and give more evaluation or prove the interpretability? I will reconsider my rating if your reply address my concern.

**Relation To Broader Scientific Literature:**

Overall, the paper situates itself within the existing body of AI and machine learning literature by extending known concepts with innovative techniques tailored for medical applications, thereby contributing both specialized knowledge and potential broader impact applications.

**Theoretical Claims:**

There is no obvious issue.

---

> ### Author Rebuttal · Authors · 2025-04-01
>
> We thank the reviewer for their detailed and constructive comments. We are especially grateful for the recognition of our contributions, including **well-supported claims** and **experiments that demonstrate a comprehensive understanding of the capabilities and limitations of the AMC method.** We address the concerns and questions raised as follows.
>
> ### [Cons1] Clinical Interpretability Evaluation
> We should emphasize that interpretableability is **not our main contribution**, and our main contributions lie in three points:
> - **Top-Down Dynamic Fusion Mechanism:** A novel top-down dynamic fusion mechanism to address the variation in data quality challenge
> - **Token Fusion for Heterogeneous Modalities:** A novel token fusion approach that can work on heterogeneous modalities and is not limited to modalities with explicit mapping relationships
> - **Extensive Real-World Validation:** Validate our method by extensive experiments on real-world datasets.
>
> #### *Interpretability and Clinical Validation*
> Although our method is **not primarily designed to enhance interpretability**, it does offer modality-level insights. Through our top-down fusion framework, we assess the importance of each modality. This helps in understanding each modality's contribution to the final prediction. Unlike traditional methods such as the Shapley value, which focus on feature-level interpretability, our approach provides insights at the modality level. Therefore, we do not directly compare our method with clinically validated medical AI methods. However, we can include specific methods in future comparisons if the reviewer suggest.
> #### *Case Study*
> To further address clinical validation concerns, we propose a case study using images from the TCGA dataset. The image is easier for readers to realize the modality data quality. This study illustrates the variation in modality importance, providing a visual representation of data quality. The results in Figure 2 of [Anonymous](https://anonymous.4open.science/r/amc_rebuttal-1247/README.md) show that images with lower modality importance scores generally exhibit poor quality, while images with higher scores indicate better quality.
>
> We will revise our introduction for clear clarification about our contribution and add the case study to our appendix.
>
> ### [Cons2] Empirical Validation for Efficiency.
>
> To address the reviewer’s concern, we report the **Mean time per iteration during training**, and **GFLOPs** during testing across baselines and AMC. The result shows our method is computationally efficient.
> |     Metric    | FlexMOE | FuseMOE | LiMOE | MAGGate |  MulT |   TF  |    AMC    |
> |:-------------:|:-------:|:-------:|:-----:|:-------:|:-----:|:-----:|:---------:|
> | Mean Time (s) |  12.73  |  18.68  | 12.65 |  11.64  | 12.85 |  12.4 |  **11.5** |
> |     GFLOPs    |  59.07  |  59.76  | 59.41 |  59.06  | 60.12 | 59.39 | **45.23** |
>
> ### [Cons3] “situates itself within the existing body of AI and machine learning literature” and has limited novelty.
> We respectfully disagree. What we really focus on advocating is the novel and practical angle of medical learning with considering multimodal data quality variance for the first time. Motivated by this fresh perspective, our solution integrates the attention mechanism and contrastive learning into the proposed token fusion framework. This token fusion approach is distinctly developed beyond existing methods. Specifically, to address previous drawbacks in token fusion, our framework offers significant differences and improvements over prior models.
>
> In addition, our novelty is consistently recognized by two reviewers. Our paper was rated as an “innovative idea“ by **Reviewer joLr**: “The proposed method is novel in how it integrates token-level rebalancing within a transformer-based model.”. Then, our novelty is also highly acknowledged by **Reviewer JTwf** as innovative ideas.
>
> Finally, our method brings novelty from the following perspectives:
> - **Novel Motivation:** we are motivated by the important observation “in medical tasks, data quality varies across different modalities and patients.”
> - **Technical Novelty:** Novel token fusion framework that does not require explicit mapping between modalities, and a novel token replacement policy when the number of modalities is larger than 2.
> - **Application Novelty:** We propose a unified learning framework that can address different kinds of medical applications: patient survival time prediction, patient mortality prediction, and diagnosis classification prediction.

---

> > ### Comment · Reviewer_CN7m · 2025-04-02
> >
> > Thanks for your reply. My primary concerns are solved and I would like to raise my rating to 3. Good luck!

---

> > > ### Author Response · Authors · 2025-04-02
> > >
> > > Dear Reviewer **CN7m**,
> > >
> > > We thank you for your thoughtful feedback and for confirming that your primary concerns have been addressed. We greatly appreciate your time and effort in reviewing our work. Please feel free to contact us if you have any further concerns.
> > >
> > > Thank you once again for all your support.
> > >
> > > Best,
> > > Authors

---

### Official Review · Reviewer_JTwf · 2025-03-10

**Overall Recommendation:** 4

**Summary:**

The paper proposes Adaptive Modality Token Re-Balancing (AMC) to address the challenge of unequal modality contributions in medical multi-modal learning. AMC dynamically quantifies modality importance and rebalances token contributions by replacing less informative tokens with inter-modal ones. It integrates differential attention, Sparse Mixture-of-Experts (SMoE), and contrastive learning to improve interpretability and efficiency. Extensive experiments on TCGA, MIMIC-IV, and ADNI datasets show that AMC outperforms existing methods, demonstrating strong generalization in multi-modal learning.

**Claims And Evidence:**

Yes

**Essential References Not Discussed:**

N/A

**Experimental Designs Or Analyses:**

The experiments are relatively solid.

**Methods And Evaluation Criteria:**

Yes

**Other Comments Or Suggestions:**

1. line 163 : “... in the modality-specific encoder to ensure the number of tokens D is the same across input modalities.”  D -> N ?
2. line 170: “We define the number of uninformative tokens for the m-th modality as Km = floor(s[m] × D).” Why is D here?
3. line 231: “level contrastive learning objective LT uses the nn-th token” nn?

**Other Strengths And Weaknesses:**

The paper is well-written, with solid experiments and innovative ideas. While it builds on existing work, the improvements lead to strong final results.

See weaknesses in the comments/questions.

**Questions For Authors:**

1. My primary concern is that in multimodal images, there often exists a misalignment issue. The proposed feature replacement and contrastive loss in the paper appear to rely on strict spatial positional constraints, which may not align with many real-world scenarios. Has the author conducted any analysis regarding the model's effectiveness in the presence of significant misalignment?
2. Frameworks based on batch processing generally facilitate testing with missing modalities. Has the author conducted tests on the performance of different modalities in the absence of certain inputs? Furthermore, how does the importance of different modalities vary across different tasks? Are there certain redundant modalities that are deemed necessary?
3. The paper lacks some intuitive visualizations. For instance, if there are corresponding examples where the token at position (2, 4) of modality A is replaced by modalities B or C, it would be beneficial to clearly demonstrate how the details of modalities B or C are more evident in this instance. Is there any intuitive visualization available to support your claims?
4. In fact, using attention mechanisms to demonstrate feature importance has become a common practice, whether in existing works on token pruning based on MLLMs, such as FastV and VisionZip, or in specific domains like EDITOR and DeMo, where attention is utilized for feature selection and MoE weighting. The author should consider comparing and citing these works to clarify their core contributions; otherwise, the overall novelty of the paper may appear insufficient.

**Relation To Broader Scientific Literature:**

The core idea of this paper is that token replacement is an interesting concept that can be applied to broader multi-modal learning tasks. It cleverly processes data along the batch dimension to avoid the computational cost of quadratic complexity while replacing tokens across different modalities during the process.

**Theoretical Claims:**

There is no clear theoretical proof in the text, and most of the content is cited from previous research findings.

---

> ### Author Rebuttal · Authors · 2025-04-01
>
> We thank the reviewer for their detailed and constructive comments. We are grateful for the recognition of our contributions, including **the interesting approach of token replacement, solid experiments, and innovative ideas.** We address the concerns and questions raised as follows.
>
> ### [Cons1] Typos
> The D in line 163 and line 170 should be N. Meanwhile the nn-th should be replaced with n-th. We will modify these typos in our revision.
>
> ### [Cons2] Misalignment Issue for the Multimodal Images.
> This is a good question. Before tokens are fed into our improved transformer blocks, we use Q-Former to encode each modality. In Q-Former, the query comprises modality-specific learnable embeddings, while the key and value derive from the input modality tokens. Q-Former extracts key information from input images, as discussed in [1], and integrates this into learnable query embeddings. Spatial information is implicitly encoded within these embeddings. **Our contrastive loss not only retains the spatial information of modalities, such as images, but also enhances the Q-Former's ability to extract critical information from image tokens.**
>
> [1] BLIP-2: Bootstrapping Language-Image Pre-training with Frozen Image Encoders and Large Language Models
>
> ### [Cons3] Modality Missing Scenario
> This paper primarily examines the contribution of different modalities to final predictions. Although handling missing modalities is not our main focus, **it is easy to extend AMC to solve the missing modality problem**. In such cases, **we treat the importance score and token score of any missing modality as zero**. This approach enables AMC to function effectively even when some data is unavailable.
> To demonstrate AMC's adaptability to the missing modality scenario, we conducted experiments using the ADNI dataset. This dataset is naturally suited for testing missing modality scenarios, providing a robust environment to validate our approach.
> The results in the following table indicate that **AMC's performance decreases only slightly when dealing with missing modalities**. This demonstrates that AMC can effectively handle the missing modality problem through simple extensions.
> We use the same model but add examples with missing modality problems. The similar performance indicates that AMC still keep prediction capability in these additional examples.
>
> | |Acc|Recall|Precision|F1|
> |-|:-:|:-:|:-:|:-:|
> | AMC| 55.91±2.49 | 55.03±2.58 | 55.38±2.55 | 54.82±2.41 |
> | AMC w/ missing modality | 56.3±2.21 | 55.77±3.32 | 57.39±2.29 | 54.81±2.18 |
>
> ### [Cons4] How does the importance of different modalities vary across different tasks?
> As shown in Figure 1, the important modality varies even for the same task across different datasets. This variation is mainly due to differences in data collection times and locations, which can affect the quality of medical data. To address the reviewer's concern, we further evaluated the single modality task performance across MIMIC-IV and ADNI (Table 1 in [Anonymous](https://anonymous.4open.science/r/amc_rebuttal-1247/README.md)). This approach helps us assess how the importance of different modalities varies across different tasks.
>
> ### [Cons5] Are there certain redundant modalities that are deemed necessary?
> In this paper, we focus on the effects of different modalities. The necessity of certain redundant modalities is beyond the scope of this work. However, this is an important question that we would like to investigate in the future.
>
> ### [Cons6] Intuitive Visualizations
> Thanks for your reminder. We provide several case studies about how each modality's tokens are replaced by other modalities’ tokens in the Figure 3 of [Anonymous](https://anonymous.4open.science/r/amc_rebuttal-1247/README.md).
>
> ### [Cons7] Comparing and Citing More Works for Novelty Certification.
> After comparing our AMC with the mentioned MLLM based methods, we summarize the following similarities and differences.
> **Similarity:**
> - All methods use the attention score to evaluate token importance.
> **Differences:**
> - Our approach uses the attention score to calculate modality importance.
> - We fuse multimodal information through our proposed token fusion framework, whereas existing MLLM methods use self-attention within the LLM backbone.
> - We consider both token-level and modality-level feature importance, unlike previous methods that focus solely on token-level importance.
>
> **Compare with More Baselines:** The input modality, medical tasks, learning objectives, and fusion framework of AMC differ significantly from those of existing methods, making it challenging to include them as baselines in our comparison.
>
> **Novelty Clarification:** Please refer to the **Cons3** in our response to the **Reviewer CN7m** for this clarification due to limited space.
>
> We are very thankful for your time and consideration. If you have any further questions, please let us know, and we will be happy to address them fully.

---

> > ### Comment · Reviewer_JTwf · 2025-04-02
> >
> > Thanks for your reply.
> > My primary concerns are solved.
> > However, in [Cons2], could you just select certain misalignment examples to explain?

---

> > > ### Author Response · Authors · 2025-04-02
> > >
> > > We appreciate your feedback and are pleased to have addressed your primary concerns.
> > >
> > > For the question in **[Cons2]**, here is our explanation:
> > >
> > > Consider an example of misalignment: an image and a sentence describing it. Assume the image contains a bird, and the word "bird" appears in the sentence. The image is divided into a token sequence $\\{M_i\\}_{i}^{N_m}$, and the text into $\\{T_i\\} _{i}^{N_t}$, where $N_m$ and $N_t$ are the counts of image and text tokens, respectively. Here, $N_m \neq N_t$, these two modalities are not aligned to each other.
> > >
> > > Our AMC uses contrastive learning as follows:
> > > - The Q-Former encodes $\\{M_i\\} _{i}^{N_m}$ and $\\{T_i\\} _{i}^{N_t}$ to $\\{x_m^i\\}_i^N$ and $\\{x_t^i\\} _{i}^{N}$ through cross-attention using modality-specific learnable queries $\\{q_m^i\\}_i^N$ and $\\{q_t^i\\}_i^N$. **$N$ is consistent across modalities to facilitate our token replacement.**
> > > - Our two-level contrastive learning objectives will apply to $\\{x_m^i\\}_i^N, \\{x_t^i\\}_i^N$.
> > > - These encoded tokens are also processed through our improved transformer blocks. Our token replacement mechanism operates on $\\{x_m^i\\}_i^N$ and $\\{x_t^i\\}_i^N$ between each adjacent transformer block layer.
> > >
> > > **The contrastive learning is applied on  $\\{x_m^i\\}_i^N$ and $\\{x_t^i\\}_i^N$ before they fed into improved transformer blocks.** Therefore, although the image and text are initially misaligned, the Q-Former and modality-specific queries $\\{q_m^i\\}_i^N$ and $\\{q_t^i\\}_i^N$ learn to align $\\{x_m^i\\}_i^N$ and $\\{x_t^i\\}_i^N$. Consequently, the contrastive loss enables the token replacement mechanism to function effectively, mitigating the misalignment issue. We also assign Figure 4 in [Anonymous](https://anonymous.4open.science/r/amc_rebuttal-1247/README.md), which shows this progress visualized.

---

### Official Review · Reviewer_jTYn · 2025-03-13

**Overall Recommendation:** 3

**Summary:**

The paper proposes Adaptive Modality Token Re-Balancing (AMC), a dynamic fusion method tailored specifically to handle varying data quality across modalities in medical multi-modal learning. AMC explicitly quantifies the importance of each modality by computing statistical information from attention distributions (mean attention values). Based on these calculated importance scores, AMC adaptively replaces less informative tokens with more valuable inter-modal tokens, guided by a two-level contrastive learning approach and an improved differential attention mechanism, thereby ensuring precise and robust integration of multi-modal medical data.

**Claims And Evidence:**

Yes.

**Essential References Not Discussed:**

Are the employed contrastive loss (Appendix A eq 7, eq 8) proposed by the authors? I believe citations are necessary.

**Experimental Designs Or Analyses:**

Which dataset is used for the experiments in Table 4&5?

**Methods And Evaluation Criteria:**

1. The method lacks a thorough analysis of its interpretability in clinical settings, which is critical for the adoption of medical AI systems.
2. The paper lacks comparisons with more recent state-of-the-art medical multi-modal fusion methods, which would strengthen its claims of superiority.

**Other Comments Or Suggestions:**

Typos: line 124, 161 n-the; line 231 nn-th token

**Other Strengths And Weaknesses:**

Section 3 is hard to follow.

**Questions For Authors:**

**1. Originality Clarification.** Token Fusion, Contrastive Loss, Differential Attention, and SMoEs are undoubtedly the most effective components driving AMC's performance gains. However, it appears that all of these components are derived from existing works.
I urge the authors to clearly **identify one contribution** that they believe most innovates or extends beyond these established techniques to demonstrate AMC's originality and impact.


**2. Choice of Modality Importance Calculation**
The authors tested various methods (Mean, STD, Max) to calculate modality importance—could you provide theoretical or intuitive justification for why the attention mean outperforms other statistics? How robust is this choice across different datasets and tasks?

**3. How does AMC perform when one or more modalities are partially or completely missing in a given patient's data?**


**4. Which dataset is used for the experiments in Table 5?**

**Relation To Broader Scientific Literature:**

The main components of AMC, including Token Fusion, Contrastive Loss, Differential Attention, and SMoEs, are adapted from existing works rather than being entirely novel.

**Theoretical Claims:**

**Insufficient Theoretical Justification**
The paper lacks rigorous theoretical justification for key design choices, such as the choice of modality importance calculation methods (Mean vs. STD vs. Max), eq 2, the replacement of indicator with eq 4.

---

> ### Author Rebuttal · Authors · 2025-04-01
>
> We thank the reviewer for the thoughtful and constructive feedback. We address the key concerns and suggestions below.
>
> ### [Cons1] Interpretability Analysis in the Medical Setting
> For the interpretable aspect, **we are not primarily aiming to propose a method that focuses on giving interpretableability**.
> However, AMC can provide modality-level insights.
> To illustrate this, we present visualizations in Figures 4 and 5.
> These figures demonstrate our model's ability to identify how each modality differently contributes to the final prediction.
> To further address reviewer concerns, we conducted a **case study** on the image modality within the TCGA dataset. Please refer to the **Cons1** in our response to the **Reviewer CN7m** for the case study details due to limited space.
>
> ### [Cons2] Recent SoTA Fusion Baselines.
> For our comparisons, we include classical fusion methods such as MAGGate, MulT, and TF. Additionally, we evaluate against recently proposed baselines: **FlexMoE and FuseMoE (both from NeurIPS 2024)**, and **MUSE (from ICLR 2024)**. The FlexMoE and MUSE are designed for medical tasks.
> If reviewers have specific baselines they recommend for inclusion, we are open to incorporating them into our analysis.
>
> ### [Cons3] Theoretical Justification about the Calculation of Modality Importances.
> As discussed in Section 5.2, Paragraph 1, we explore two perspectives to calculate modality importance: **the statistical perspective** (Mean v.s. STD) and **the value perspective** (Mean v.s. Max) of attention
> In this paper, we propose a novel fusion mechanism to address the challenge of modality quality variation. Our empirical evidence supports the effectiveness of using Mean for determining modality importance. Although this paper does not include a theoretical analysis, our contributions have significant practical implications. Rigorous proof is valuable, but it is outside the scope of this work.
>
> ### [Cons4] Dataset in Table 4&5
> Thanks for pointing this, the datasets used in Table 4&5 are the MIMIC-IV. We will revise the table captions for better standalone clarity.
>
> ### [Cons5] Citation about Contrastive Learning
> Thanks for the reminder. We will include the related citations [1], [2] for instance level and token level contrastive learning respectively in our revision.
>
> [1] BLIP: Bootstrapping Language-Image Pre-training for Unified Vision-Language Understanding and Generation
> [2] Token-Level Contrastive Learning with Modality-Aware Prompting for Multimodal Intent Recognition
>
> ### [Cons6] Novelty Clarification
>
> Please refer to the **Cons3** in our response to the **Reviewer CN7m** for this clarification due to limited space.
>
> ### [Cons7] Section 3 is Hard to Follow
> Section 3 introduces token fusion and discusses its drawbacks. The basic idea of token fusion is to prune single-modal transformers and reuse pruned units for multimodal fusion. Each single-modal transformer is pruned individually, and pruned units are replaced by **projected alignment features** from other modalities.
>
> The drawbacks include: (1) **the requirement for well-aligned modalities with explicit mapping**, (2) **the need to manually set a threshold $\theta$ for pruning, which requires additional effort to determine a feasible value**, and (3) when the number of modalities exceeds two, tokens are randomly selected from other modalities as substitutes, **assuming all modalities capture the correct information.** We are open to refining this section further if the reviewer provides specific comments.
>
> ### [Cons8] Typos
> Thanks for the feedback. We will correct these typos in our next revision.
>
> ### [Cons9] The Intuitive Justification and Robustness Clarification of Modality Importance Calculation
> **Intuitive Justification:** For a reliable modality, attention values on each token should **vary to capture meaningful information**. At the same time, **the overall attention value should be high** to ensure that useful information is integrated effectively. This suggests that the attention map captures meaningful information across the input data. Both standard deviation (STD) and maximum (Max) can be affected by outliers. In contrast, a higher mean value is more robust.
>
> **Robustness Clarification:** We conducted an ablation experiment on the ADNI dataset to assess robustness. The F1 scores for STD and Max were 53.93 and 52.99, respectively, which are lower than the F1 score of 54.82 achieved by Mean. This demonstrates that robutness of Mean operation for modality importance.
>
> ### [Cons10] Missing Modality Scenarios
> Please refer to the **Cons3** in our response to the **Reviewer JTwf** for this clarification due to limited space.
>
> We are very thankful for your time and consideration. Please let us know if you have any further questions; we will be happy to address them fully.

---

### Official Review · Reviewer_joLr · 2025-03-19

**Overall Recommendation:** 3

**Summary:**

This paper proposes Adaptive Modality Token Re-balancing (AMC), a transformer-based fusion method that dynamically adjusts token contributions from different modalities based on estimated modality importance scores derived from attention distributions. AMC selectively replaces less informative tokens from weaker modalities with more informative ones from stronger modalities. The approach also integrates a differential attention mechanism, a Sparse Mixture-of-Experts (SMoE) layer to address inter-modal gradient conflicts, and a two-level contrastive loss for feature alignment. Although each component individually builds upon existing methods, their combined application specifically targets modality imbalance in medical AI. The method demonstrates improved interpretability, generalizability, and predictive accuracy on three multimodal medical datasets, TCGA (cancer survival prediction), ADNI (Alzheimer’s disease classification), and MIMIC-IV (hospital mortality prediction), and one non-medical benchmark (ENRICO), outperforming state-of-the-art multimodal fusion methods such as FlexMOE, LiMOE, MAGGate, MulT, TensorFusion, and MUSE.

**Claims And Evidence:**

The paper makes three primary claims:

First, the authors assert that explicitly recognizing modality contributions, as done by AMC, offers advantages over previous implicit methods. However, AMC computes modality importance explicitly using self-attention scores, which themselves are implicitly learned and subject to dataset biases. Since token replacement is guided by these scores without external validation (e.g., human-labeled importance or uncertainty measures), the claim of explicit modality contribution recognition is overstated and requires clarification.

Second, the authors highlight improved interpretability as a major contribution. However, this interpretability remains limited to computational visualization of modality importance and token reallocation. The paper lacks clinical validation, such as alignment with clinical decisions, biological plausibility, or qualitative assessments by domain experts. Comparisons to clinically validated explainable medical AI methods are also absent, limiting the strength of this claim.

Third, the authors claim the Sparse Mixture-of-Experts (SMoE) layer effectively manages gradient conflicts between modalities. Although numerical results indicate performance drops when removing SMoE, no direct evidence or analysis demonstrates how SMoE resolves gradient conflicts, particularly from a clinical perspective.

**Essential References Not Discussed:**

The paper focuses exclusively on transformer-based approaches and does not discuss alternatives.

**Experimental Designs Or Analyses:**

The authors did not conduct a failure case analysis, leaving it unclear under what conditions and why AMC might fail. For example, AMC is not the best-performing model for certain cases, such as the LGG cancer type in TCGA, yet the authors do not investigate or discuss possible reasons. Additionally, since AMC's re-balancing process relies entirely on self-attention distributions, which can be unreliable indicators of feature importance, biased or inconsistent attention scores might cause the token replacement mechanism to amplify errors instead of reducing them.

**Methods And Evaluation Criteria:**

The paper evaluates AMC on diverse medical datasets (TCGA, ADNI, MIMIC-IV), which are widely used benchmarks in multimodal medical research. However, real-world medical datasets often suffer from missing modalities. AMC assumes all modalities are consistently available, limiting its practicality for real-world clinical deployment.

**Other Comments Or Suggestions:**

1. While the paper emphasizes data quality variations, it does not formally define or objectively measure "quality," relying instead on performance differences and attention scores as proxies. An explicit metric or clearer definition would enhance clarity.

2. Figure 3, illustrating the customized token fusion workflow, could be presented more clearly.

**Other Strengths And Weaknesses:**

Strengths: The proposed method is novel in how it integrates token-level rebalancing within a transformer-based model. The reproducibility is reasonable. Experiments are reasonably thorough in terms of dataset selection and metric evaluation.

Weaknesses: The paper lacks discussion of computational costs, specifically how token rebalancing affects training or inference time compared to existing fusion methods. Absence of error analysis, failure-case examination, and handling of missing modalities limits AMC's practical value as a real-world medical AI solution.

**Questions For Authors:**

1. How does AMC handle missing modalities? If a modality is unavailable (e.g., missing genetic data in TCGA), does AMC fail, or can it adapt accordingly?
2. Can the authors provide an analysis or insights into scenarios where AMC underperforms or fails, and explain the reasons behind such outcomes?
3. How do the authors quantitatively demonstrate that the SMoE layer effectively resolves inter-modal gradient conflicts? Additionally, why was SMoE chosen over alternative gradient stabilization techniques?
4. The paper discusses efficiency claims qualitatively but lacks quantitative evaluation. Could the authors provide specific metrics regarding AMC’s computational efficiency?

**Relation To Broader Scientific Literature:**

N/A

**Theoretical Claims:**

N/A

---

> ### Author Rebuttal · Authors · 2025-04-01
>
> We thank the reviewer for their time and thoughtful feedback. We are encouraged that the reviewer finds our method **novel in how it integrates token-level rebalancing** with **reasonable reproducibility and evaluation**. Below, we address each concern in turn.
>
> ### [Cons1] Overstated About the Modality Importance
>
> To respond to the reviewer's concern regarding the importance of modalities, we propose two experiments to clarify the role of modality scores in identifying the significance of each modality.
> 1. **Dataset Level Modality Importance:** We assess modality importance at the dataset level using the MIMIC-IV dataset. Each modality is evaluated individually using the same AMC backbone. The results in the following table indicate that the 'Code' modality is less important (the lowest F1, Recall, and Precision), aligning with Figure 4 where 'Code' have a higher frequency of token replacement.
>
> |Modality|  Code |  Note |  Lab  |
> |--------|:-----:|:--:|:----:|
> | Acc       | 67.95 | 67.58 | 64.77 |
> | Precision | 55.44 | 55.82 | 59.18 |
> | Recall    | 55.88 | 56.51 | 62.50 |
> | F1        | 55.60 | 56.02 | 58.67 |
>
> 2. **Instance Level Modality Measurements:** We investigate the effectiveness of our modality score in reducing prediction uncertainty. We compare the uncertainty of predictions ($-\sum_{i=1}^{N}p_i\log(p_i)$, where $p_i$ is the predicted probability of class $i$) using AMC with its original modality scores against AMC with equal modality scores. Our findings show that the modality importance score effectively reduces prediction uncertainty, demonstrating its utility in identifying significant modalities.
>
> |                                    | Uncertainty |
> |------------------------------------|:-----------:|
> |                 AMC                |    0.1218   |
> | AMC w/ average modality importance |    0.1343   |
>
> These experimental results indicate that our modality importance adequately captures modality contribution for the final prediction.
>
> ### [Cons2] Clinical Validation of the Interpretableability of AMC
>
> Please refer to the **Cons1** in our response to the **Reviewer CN7m** for this clarification due to limited space.
>
> ### [Cons3] Extend to Missing Modality Scenario
>
> Please refer to the **Cons3** in our response to the **Reviewer JTwf** for this clarification due to limited space.
>
> ### [Cons4] The Measurement of Modality Quality Definition
>
> **We define the modality quality as the performance of each modality with the same modal backbone.** As we show in Figure 1, the bottom part shows the performance variations with the same network backbone but different modalities as the input. **These performance differences of each modality across different cancer types show the challenge of data quality variation.**
>
> ### [Cons5] Gradient Conflict between Different Modalities.
> We analyze the distribution of cosine distances between training gradients derived from the different modalities in the MIMIC-IV dataset. These gradients are computed from a model configured with experts and a dense MLP, both following the same configuration as in the AMC and Dense Model setups. The gradients are extracted from the final transformer layer.
> **Higher positive cosine distances indicate reduced gradient conflict**, which suggests better alignment between the modalities in terms of learning direction.
> Our result in Figure 1 of [Anonymous](https://anonymous.4open.science/r/amc_rebuttal-1247/README.md) shows that the SMoE effectively alleviates the gradient conflict between different modalities.
>
> ### [Cons6] Training Efficiency
> Please refer to the **Cons2** in our response to the **Reviewer CN7m** for this clarification due to limited space.
>
> ### [Cons7] Illustrating the customized token fusion workflow of Figure 3.
> Thanks for pointing that out.
> In our token fusion, we first identify the token score (*Equation 3*) and modality importance (*Equation 4*) via the attention distribution. Then we utilize this information to decide which token should be replaced and selected to impute intra-modal tokens (*Equation 4*). Finally, we use $\{{x}_{m}\}_{m=1}^{M}$ for the final prediction. In the revision, we will revise the figure captions for better standalone clarity.
>
> ### [Cons8] Fail/Underperform Analysis of AMC
> When **all modalities possess high quality**, determining which token to replace in AMC's token fusion process becomes challenging. This can lead to potential information loss due to token replacement, which could explain why AMC underperforms compared to MulT and MAGGate in the LGG cancer type of the TCGA dataset.
> Despite this, AMC still achieves a top-3 performance among seven baseline methods in the TCGA LGG dataset. This demonstrates that while AMC may not be the best performer in this specific case, it still maintains competitive performance overall.
>
> We sincerely thank you for your time and consideration. Please let us know if you have any further questions; we are more than happy to address them comprehensively.

---

### Decision · Program_Chairs · 2025-05-01

**Decision:**

Accept (poster)

**Comment:**

This paper proposes Adaptive Modality Token Re-balancing (AMC), a transformer-based fusion method that dynamically adjusts token contributions from different modalities based on estimated modality importance scores derived from attention distributions.

Reviewers generally appreciated the novel idea of token-level rebalancing within a transformer architecture, and recognised the solid experimental results supporting the proposed method. The rebuttal was also helpful in clarifying several technical concerns, particularly around modality misalignment handling and robustness to missing inputs.